# Effect of Preventive Exercise Programs for Swimmer’s Shoulder Injury on Rotator Cuff Torque and Balance in Competitive Swimmers: A Randomized Controlled Trial

**DOI:** 10.3390/healthcare13050538

**Published:** 2025-03-01

**Authors:** Nuno Tavares, João Paulo Vilas-Boas, Maria António Castro

**Affiliations:** 1Faculty of Sports, University of Porto and CIAFEL, 4200-450 Porto, Portugal; 2RoboCorp, I2A, Polytechnic Institute of Coimbra, 3045-093 Coimbra, Portugal; maria.castro@ipleiria.pt; 3Centre of Mechanical Engineering, Materials and Processes (CEMMPRE), University of Coimbra, 3030-788 Coimbra, Portugal; 4LABIOMEP-UP—Porto Biomechanics Laboratory, Faculty of Sports and CIFI2D, University of Porto, 4200-450 Porto, Portugal; jpvb@fade.up.pt; 5School of Health Sciences, ciTechCare, CDRSP, Polytechnic University of Leiria, 2411-901 Leiria, Portugal

**Keywords:** swimmer’s shoulder injury, elastic band program, weight program, rotator cuff, injury prevention, preventive medicine

## Abstract

Background: Over the season, competitive swimmers experience a progressive imbalance in rotator cuff strength, predisposing them to a significant risk factor for a swimmer’s shoulder injury. Objectives: Verify the effectiveness of two 12-week preventive programs on the shoulder rotators’ peak torque and conventional/functional ratios. Design: A care provider- and participant-blinded, parallel, randomized controlled trial with three groups. Participants: Competitive swimmers aged 16 to 35 years with no prior clinical issues related to their shoulders. Interventions: Twice a week, over 12 weeks, the two experimental groups performed five exercises where the only difference was executing the program with weights or elastic bands, and the control group performed a sham intervention. Main outcome measures: The concentric and eccentric peak torque of the internal and external rotators of the dominant shoulder were assessed before and after the intervention using an isokinetic dynamometer Biodex System 3, at 60°/s, 120°/s, and 180°/s. Results: Among the experimental groups, only one test indicated a reduction (*p* ≤ 0.05) in rotator peak torque, while the control group showed a decrease (*p* ≤ 0.05) in five tests. Swimmers who completed the prevention programs demonstrated less imbalance in conventional/functional ratios than controls. Conclusions: Implementing a 12-week preventive program minimizes the progressive shoulder rotational imbalance over the season in competitive swimmers. Clinical Trial Registration number: NCT06552585.

## 1. Introduction

Swimmer’s shoulder is the most common injury among swimmers [1,2,3,4,5,6]. It is characterized by non-specific anterior shoulder pain resulting from the repetitive impingement of the rotator cuff under the coracoacromial arch [3,7]. This impingement can result in functional impairments and decreased athletic performance [8]. It is estimated that around 40–91% of swimmers will experience this issue at least once during their careers [2,6,9,10]. Furthermore, 20–35% of competitive swimmers experience a loss of training or competition time each year due to this injury [9,11]. The causes of swimmer’s shoulder are dynamic and multifactorial [5,7,12]. Several risk factors contribute to this painful syndrome, including muscle strength imbalances [2,3,8,13], swimming volume [1,8,12,13], fatigue [12,13,14], competitive level [8,13], poor stroke mechanics [13,15], lack of scapular stability [3,8,13,16], changes in mobility [2,3,8,13,16], and a previous history of injury [1,8,13].

Modifiable risk factors are crucial for identifying athletes at increased injury risk. Many of these factors can be altered through therapeutic interventions [5]. Rotator cuff imbalances are a modifiable musculoskeletal risk factor and one of the strongest predictors of swimmer’s shoulder [7,13]. These muscle imbalances can lead to abnormal movements of the glenohumeral and scapulothoracic joints, which may continuously aggravate susceptible tissues [14]. Usually in overhead sports athletes, the external rotation (ER) and the internal rotation (IR) peak torques (PTs) in a concentric or eccentric action, measured through the isokinetic dynamometer [9,17,18,19], are the main outcomes collected to evaluate this risk factor [9,17,20]. Through these values, it is possible to calculate the strength balance of shoulder rotators using different ratios [9,19,20]. The conventional ratios are the division between ER and IR PTs in a concentric (conER:conIR) or eccentric (eccER:eccIR) contraction mode. The functional ratios are calculated by dividing the ER PT during an eccentric action by the IR PT during a concentric action (eccER:conIR), or the reverse (eccIR:conER). The eccER:conIR ratio is generally deemed more suitable for evaluating the eccentric action of the antagonist muscles, which contributes to the dynamic stability of the glenohumeral joint. Conversely, the eccIR:conER ratio reflects the arm cocking action during overhead throwing [9].

Several studies have verified reference ratio values to predict or not predict shoulder pain associated with sports [21,22,23,24,25]. However, some controversy remains regarding the appropriate ratio intervals for competitive overhead athletes [17]. Ellenbecker and Davies [24] concluded that to prevent shoulder injuries in overhead sports the conER:conIR ratio should be between 0.66 and 0.75. Consequently, the strength of the ER should be 2/3 compared to the IR of the shoulder. Regarding functional ratio, Bak and Magnusson [22] reported an eccER:conIR ratio = 0.86 in asymptomatic swimmers. Drigny et al. [9] considered that swimmers have a 4.5-fold increased risk of in-season shoulder injury with an eccER:conIR ratio <0.68.

Due to many training and competition events, there is a normal tendency during the season for a progressive imbalance of shoulder rotator ratios in competitive swimmers [20]. Batalha et al. [17], found a decrease from 0.79 to 0.71 in the conER:conIR ratio in swimmers after 8 months of a competitive season. Strength prevention programs have been tested to minimize this predisposition for imbalance during a swimmer’s season and normally show good results [5,7,26,27,28,29,30,31]. However, some authors report no effect of these preventive programs on strength and endurance variables in swimmers [32,33] and other overhead sports athletes [34]. In general, these studies are characterized by a big heterogeneity: in the quantity and type of exercises performed during the program, in the kind of instrument used in the exercises, and in the monitoring, duration, and progression of the program over time. Therefore, it does not allow us to draw strong conclusions regarding the real effect of each preventive program on this musculoskeletal risk factor for the swimmer’s shoulder injury. Recent reviews indicate that strength programs with a few exercises performed out of the water and in an open kinetic chain appear to have better results in preventing this sports injury [5,7], pointing to the need to clarify precisely the dimension of this effect.

Verifying the difference between performing open kinetic chain exercises with weights or elastic bands is important. In shoulder exercises, the elastic band typically produces more electromyographic activity in the posterior muscles, while weights engage the lateral and anterior muscles [35]. This significant distinction may lead to different potential effects from each exercise program.

Furthermore, it is necessary to check how long the preventive program should be to produce an effect on the strength balance of the rotator cuff. Some studies indicated no effects of the 8-week [36] or 6-week preventive programs used [32,33]. According to the American College of Sports Medicine guidelines, strength or endurance exercise programs demonstrated significant improvements only after 10 weeks of training [36].

Additionally, there is a need to clarify the impact of individual monitoring of the program execution. Some tested programs do not indicate monitoring [26,37] or seem to do group monitoring [32].

The primary aim is to verify the effectiveness of two 12-week preventive programs for swimmers’ shoulder injuries on the IR and ER PTs and the respective conventional and functional ratios. These programs include regular monitoring and progression over time. One program utilizes weights while the other employs elastic bands. This objective seeks to clarify the lack of consensus on the most effective preventive programs. The secondary aims include examining the differences between the weight and elastic band programs, evaluating whether a 12-week program duration leads to significant results, and analyzing the impact of individualized monitoring on reducing dropout rates during the follow-up period. The study hypothesizes that the two tested preventive programs reduce rotator cuff imbalances in competitive swimmers. Conversely, the absence of a preventive program does not affect rotator cuff strength and balance.

## 2. Materials and Methods

### 2.1. Trial Design

A care provider- and participants-blinded, parallel, randomized controlled trial with three groups was completed from September 2022 to April 2023. This trial follows the CONSORT 2010 checklist for reporting randomized controlled trials (Appendix A). All procedures received approval from the Ethics Committee of the Polytechnic Institute of Coimbra (CEIPC 6/2022). The protocol for the randomized controlled trial has been registered on ClinicalTrials.gov with the registration number NCT06552585.

### 2.2. Participants

Swimmers were recruited from two national competitive teams in Portugal, affiliated with the Portuguese Swimming Federation, during the 2022/2023 season. To be included in the study, participants had to be aged between 16 and 35 years, have at least five years of experience in national competitions [26], and engage in a minimum of eight hours of swimming training per week [26,32,38]. Exclusion criteria included a history of shoulder pain within the past six months [28,32,39], prior shoulder surgeries [27,29,30,39], traumatic shoulder injuries, cervical or thoracic conditions [39], range of motion (ROM) deficits, or neurological injuries [40,41,42]. Exclusion also applied to swimmers undergoing recent treatment, such as physical therapy, injections, or medications, for at least six weeks [27,29]. All eligibility criteria were verified through an individual questionnaire before sample selection. Each participant read and signed a written informed consent form based on the revised version of the Declaration of Helsinki from 2013 [43]. After selection, all swimmers were weighed using a Tanita RD-953 body composition scale, and their height was measured with a stadiometer. The assessment of female athletes was conducted outside of their menstruation period. The confidentiality of participants’ identities was maintained following the European Union General Data Protection Regulation (Regulation (EU) 2016/679). This randomized controlled trial was conducted at the RoboCorp Laboratory at the Polytechnic Institute of Coimbra.

### 2.3. Intervention Programs

Twice a week for 12 weeks [30,44], two intervention groups participated in a preventive program consisting of five open kinetic chain exercises commonly recommended in the literature for preventing swimmer’s shoulder injury [7,26,27,28,29,30,31,32,33,37,38,39]: internal rotation (IR) at 90°, external rotation (ER) at 90°, scapular punches, T’s, and Y’s (Table 1). The weight program group performed these exercises using Domyos weights of 1, 2, 3, 4, or 5 kg (Figure 1), while the elastic band program group used Bodytone Power elastic bands with resistance levels of 10, 15, 20, 25, or 30 kg (Figure 2). The load for each participant was assessed and adjusted to 75% of their one-repetition maximum (1RM) [44]. After six weeks of training, each swimmer underwent another 1RM test to evaluate their ability to progress the load [37,39]. The protocol for the 1RM test followed the guidelines outlined in Baechle and Early [44]. The T’s exercise was selected for this assessment. Before the 1RM test, a warm-up consisting of ten repetitions of the T’s exercise was performed using a very light load, either with weights or an elastic band. For each preventive exercise, participants completed two sets of ten valid repetitions [44]. Repetitions were considered valid when the swimmer achieved each exercise’s target ROM, as detailed in Table 1. Each repetition consisted of five-second concentric and eccentric phases [40]. A Tabata Timer mobile application was used to monitor this timing. Additionally, one minute of rest was allowed between different exercises and sets [44]. The programs were supervised by a sports physiotherapist who was independent of the study. Each athlete received personalized monitoring, which included regular corrections to their exercise technique during all sessions. The first session was conducted in person, while the remaining sessions were monitored online via WhatsApp Messenger or Zoom Video Communications, Inc. Each session was individualized and lasted about 30 min.

### 2.4. Control Group

The control group carried out a sham intervention twice a week for 12 weeks. This intervention consisted of two sets of ten repetitions of five shoulder mobility exercises, commonly included as part of a warm-up before training. The sham intervention exercises were shoulder maximum flexion and extension, horizontal abduction and adduction starting from 90° of shoulder abduction, maximum IR and ER starting from 90° of shoulder abduction, circumduction of the shoulder in a clockwise direction, and circumduction of the shoulder in a counterclockwise direction [44]. There was no progression in the exercises over time. While the team’s coach monitored the execution of the exercises, there was no individualized oversight or regular technique correction provided by a physiotherapist.

To ensure ethical equity among the sample groups and considering the potential benefits of the exercise programs tested, the control group was guaranteed access to either the weight or elastic band program after the conclusion of the trial.

### 2.5. Outcomes and Testing Procedures

Before (T0) and after 12 weeks (T1) of interventions and control procedures, the concentric and eccentric peak torque (PT) of IR and ER of the dominant shoulder was assessed through an isokinetic dynamometer Biodex System 3 (Biodex Medical Systems, New York, NY, USA) [26,38,45,46,47], at 60°/s, 120°/s, and 180°/s [37].

Previously to strength evaluations, each swimmer completed a 10 min warm-up that involved general upper limb joint mobilization [45]. The isokinetic dynamometer assessment was performed with the swimmer lying [46,47]. The upper extremity was positioned with the shoulder abducted to 90° and the elbow flexed to 90° [47], ensuring that the dynamometer’s axis was aligned with the longitudinal axis of the humerus [46]. The chosen evaluation position is similar to the body position of athletes during swimming. This position demonstrates better reproducibility and reliability for assessing the IR, ER, and concentric conventional ratio compared to both seated assessment and lying evaluation with the shoulder at 45° of abduction [46,48]. Forthomme et al. [48] conducted strength assessments in this position at angular velocities of 60°/s and 240°/s showed reproducibility values of 11.8 and 10.5 for IR, 8.9 and 7.5 for ER, and 7.6 and 7.8 for the conventional concentric ratio. The reliability values were noted as 15.9 and 15.5 for IR, 9.6 and 6.6 for ER, and 0.15 and 0.13 for the conventional concentric ratio. Velcro straps were used to stabilize the athlete’s trunk and the evaluated arm to prevent compensatory movements (Figure 3) [26,45,46]. Gravity correction was not applied during this testing position because the ER and IR shoulder muscles moved with and against gravity when applying force [46,47,49]. Strength was tested through a ROM of 145°, measured from 80° of ER to 65° of IR, to capture the PT variation within the maximum available rotational ROM. The tests began with the shoulder positioned at 80° of ER [46]. Concentric strength was tested first, followed by the eccentric strength test [21,47]. The order of the angular velocity test was increased (60°/s–120°/s–180°/s) to preserve an ascending difficulty during the evaluations [46]. A hard cushion deceleration control was chosen to give the subjects the greatest potential for achieving maximum velocity before deceleration [46,50]. Before testing, the procedure was explained to all athletes, emphasizing the importance of exerting maximal effort within their tolerances [45]. Subsequently, swimmers performed three submaximal trials to familiarize themselves with the ROM and the accommodating resistance of the dynamometer [47]. Each of the concentric and eccentric tests at varying velocities involved five maximal-effort reciprocal repetitions [46,47]. Standardized verbal instructions and encouragement were provided to all subjects [45,46,47], and a one-minute rest period was maintained between different tests [38].

The PT is the maximum torque produced at any point during the ROM and was used to assess the rotator cuff strength [26]. To evaluate the rotator cuff strength balance, the ER:IR ratio was calculated. This ratio is obtained by dividing the ER PT by the IR PT and multiplying the result by 100 [26,47]. For this study, the conventional concentric ER: concentric IR (conER:conIR) and functional eccentric ER: concentric IR ratio (eccER:conIR) were calculated.

### 2.6. Sample Size

The sample size calculation was performed using G*Power software (Franz Faul, Edgar Erdfelder, Axel Buchner, Universität Kiel, Kiel, Germany, version 3.1.9.4). The statistical test selected was designed to compare mean differences between independent groups. The effect size used for this calculation was derived from a similar clinical trial [26], which involved 56 participants divided into three groups: an experimental group of 20 swimmers who participated in both a preventive strength program and swimming training, a training group of 20 swimmers who only engaged in swimming training, and a control group consisting of 16 active non-swimmers. Based on this trial [26], an effect size (Cohen’s d) of 2.81 was identified, representing the smallest difference considered statistically significant between the experimental and training groups when assessing IR and ER PT at 60°/s and 180°/s of the swimmers’ dominant shoulder.

Using an alpha level of 0.01, a power of 0.99, and an allocation ratio of 1:1, the software projected a minimum total sample size of 24 swimmers (8 per group). To account for potential dropouts, a final sample size of 30 swimmers (10 per group) was considered.

### 2.7. Randomization and Blinding

The initial contact with the competitive swimmers of the two swimming teams was made through a telephone call, during which the eligibility criteria and individual consent for participation in the trial were screened. The eligible swimmers were then divided into three groups: a weight program group, an elastic band program group, and a control group. Allocation to each group was performed through stratified randomization according to the team (A or B), sex (male or female), and main swimming style (butterfly, backstroke, breaststroke, front crawl ≤ 200 m, or front crawl > 200 m). This approach aimed to balance the potential effects of these three variables on the results [27]. Participants were initially screened to determine their respective strata. The allocation process was concealed within each stratum, employing a computer-generated random number managed by a member of the research team. The participants were unaware of the existence of other sample groups. Then, all swimmers carried out the T0 assessment through the isokinetic dynamometer. To minimize fatigue bias during the assessment, half of the swimmers started with their dominant shoulder, while the other half began with their non-dominant shoulder. This order was randomized using a computer-generated random number. Following the T0 assessment, two 12-week experimental interventions were implemented and supervised by a sports physiotherapist who was external to the study and unaware of the previous procedures or the study’s objectives. The first session with each athlete was conducted in person. During this session, the physiotherapist performed an individual 1RM test according to the specific equipment and explained the exercise program to the swimmer. After 6 weeks of intervention, the physiotherapist conducted a new 1RM evaluation to check any potential progress in load. All in-person and online sessions were individualized to ensure that each athlete remained unaware of the other groups. The sports physiotherapist also tracked the number of sessions completed by each athlete and did not have contact with those in the control group. At the end of the 12 weeks, all swimmers who completed the full set of proposed interventions carried out the T1 strength assessment through the isokinetic dynamometer. All T0 assessment procedures were maintained in the T1.

### 2.8. Data Analysis

The data were previously filtered (smoothing option) and windowed at 95% of test velocity. Data analysis was performed using the Acqknowledge 4.1 software. The mean and standard deviation of the PT in the five repetitions of different muscles (IR and ER), actions (concentric and eccentric), and angular velocities (60°/s, 120°/s, and 180°/s) were calculated. Subsequently, the conventional conER:conIR and the functional eccER:conIR ratios were calculated for three angular velocities.

All statistical analysis was carried out using IBM SPSS Statistics 27 software. Mean and standard deviation were used for sample characterization. The Shapiro–Wilk test was used to verify the normality of the sample distribution. The homogeneity of variance between the three groups was verified for the sample characterization variables and the IR and ER PT at T0, through the one-way ANOVA or the Kruskal–Wallis tests. After this, an intra-group analysis between T0 and T1 was made. The paired sample t-test or the Wilcoxon test was applied to compare differences in each group between T0 and T1. A *p*-value ≤ 0.05 was considered the significance level for each difference in PT, conventional, and functional ratios.

## 3. Results

Participant recruitment happened in September 2022. Initial contact with the competitive swimmers was made through a phone call, during which the eligibility criteria and individual consent for participation in the trial were screened. Out of 49 athletes contacted, 30 were eligible and deemed suitable to participate. These participants were allocated through stratified randomization into three groups: a weight training program group (n = 10), an elastic band training program group (n = 10), and a control group (n = 10). The different types of swimming training according to the team or the main swimmer’s style and the strength differences between males and females were two variables that could influence the results. To minimize its impact, each group maintained equal representation, consisting of five swimmers from team A and five from team B, five men’s and five women’s, and two swimmers from each swimming style—butterfly, backstroke, breaststroke, front crawl ≤ 200 m, and front crawl > 200 m (Figure 4). The three groups exhibited homogeneity of variance (*p* > 0.05) regarding sample characterization (Table 2), and in internal rotation (IR) and external rotation (ER) peak torque (PT) at T0 (Table 3). Each group performed two strength assessments: one before the intervention (T0) and one after the intervention (T1). No participant dropouts were recorded during the 12 weeks of follow-up. All participants in the experimental interventions completed the 24 individual sessions of their prevention program over 12 weeks.

### 3.1. Peak Torque

Firstly, an intra-group analysis of PT variation was conducted between T0 and T1 (Table 4). Overall, there was a decrease in PT values between the two moments. In the weight program group, there was a decrease in eight PT assessments, while the remaining groups exhibited decreases in nine assessments. A total of six differences (*p* ≤ 0.050) were found in the PT values between T0 and T1, all indicating a drop in values. For the intervention groups, only one difference of −5.65 Nm/kg in concentric IR PT at 60°/s was considered significant (*p* = 0.047) in the weight program group. The remaining five differences were observed in the control group, which included: concentric IR PT at 60°/s (*p* = 0.002), eccentric IR PT at 60°/s (*p* = 0.036), eccentric ER PT at 60°/s (*p* = 0.048), concentric IR PT at 120°/s (*p* = 0.005), and concentric IR PT at 180°/s (*p* = 0.002).

### 3.2. Conventional and Functional Ratios

The concentric ER: concentric IR (conER:conIR) and eccentric ER: concentric IR (eccER:conIR) ratios were calculated for the different assessments at T0 and T1. Each swimmer provided three values for the conventional and functional ratios during each evaluation, corresponding to the three angular velocities tested. These individual values were plotted in three graphs for each group (Figure 5, Figure 6 and Figure 7), which also marked reference values for preventing shoulder injuries [9,22,24]. According to Ellenbecker and Davies [24], the optimal range for the conER:conIR ratio in swimmers is between 0.66 and 0.75. Regarding the eccER:conIR ratio, the optimal range is between 0.68 and 0.86, based on findings from Bak and Magnusson [22] and Drigny et al. [9].

The weight program group showed an overall increase in the conER:conIR and eccER:conIR ratios from T0 to T1. This increase resulted in a higher number of T1 assessments in the optimal range of the conER:conIR ratio but a smaller number of T1 evaluations in the optimal range of the eccER:conIR compared to T0 (Figure 5).

In the elastic band program group, there was an overall increase only in the conER:conIR ratio from T0 to T1. This increase resulted in fewer T1 assessments in the optimal range of the conER:conIR ratio compared to T0. Additionally, there was no significant change in the eccER:conIR ratio between T0 and T1 (Figure 6).

In the control group, there was an increase in T1 compared to T0 for both shoulder ratios. This increase causes a move away from the optimal ranges of the conER:conIR and eccER:conIR ratios (Figure 7).

In summary, a more significant decrease in the shoulder internal rotation peak torque values in the control group increased the conventional and functional ratios, moving these values further away from the non-injury zone, compared to the experimental groups.

## 4. Discussion

This investigation primarily aimed to assess the effectiveness of two 12-week preventive programs for swimmers’ shoulder injuries, focusing on the internal (IR) and external rotation (ER) peak torques (PTs), as well as the respective conventional and functional ratios. The findings indicated that the absence of a preventive exercise program resulted in significant decreases in PT values for both internal and external rotators, with particularly notable declines in the concentric assessment of internal rotators. Regarding rotator cuff balance, the results showed that the experimental groups demonstrated less imbalance compared to the control group.

Regarding secondary objectives, neither preventive program was more effective than the other. Both exercise programs successfully minimized the typical decrease in PT values during the swimming season. The weight program seemed to enhance the conventional concentric ratio, while the elastic program maintained the functional ratio, which includes the eccentric component. Additionally, a duration of 12 weeks for the programs seems to be sufficient to detect some effects. Lastly, individual monitoring throughout this period proved to be an effective strategy for reducing dropout rates and improving adherence to the program.

In the intra-group analysis, no significant increases were observed in any PT assessments following the experimental or control procedures. Instead, there was a general trend of decreasing PT values. This finding is consistent with some previous studies [32,33,34]. This decline may be attributed to fatigue resulting from the cumulative effects of swimming training and competitions throughout the season. Some authors have observed a reduction in shoulder IR PT due to fatigue in other overhead sports immediately after training [51] and following several weeks of training [34]. In contrast, some studies have reported an increase in shoulder ER PT among competitive swimmers after completing a preventive open kinetic chain program, as measured with an isokinetic dynamometer [26,37,38]. The variety of exercises included in these programs could explain this difference. Notably, two of these programs [37,38] incorporated more than one exercise that specifically emphasized ERs, which could account for better outcomes. Conversely, our experimental programs included only one exercise targeting shoulder ER–ER at 90°. Furthermore, it is important to highlight those five of the six significant decreases in shoulder PT occurred in the control group. This suggests that experimental procedures may help minimize the expected declines in rotator cuff PT throughout the swimming season.

Regarding rotator cuff balance, swimmers who participated in experimental prevention programs demonstrated less imbalance after 12 weeks. The most significant differences between the initial (T0) and the final (T1) assessment were observed in the control group, which showed a marked deviation from the injury prevention ratios in both conventional and functional ratios. The experimental groups exhibited deviations in only one of the ratios. Specifically, the weight training group showed a deviation in the functional ratio, while the elastic band training group in the conventional concentric ratio. Batalha et al. [17] and Drigny et al. [9] observed that throughout the swimming season, there is a tendency for changes in swimmer’s rotational ratios for injury risk values. Additionally, Fredriksen et al. [34] verified that in another overhead sport, training for 8 weeks increased conventional and functional ratios.

The qualitative analysis of rotator cuff balance involved comparing conventional and functional ratio values against reference ranges for preventing shoulder injuries in athletes. While these ranges are widely cited in the literature [9,24,38], they have some limitations. The recommended range of 0.66–0.75 for the concentric conventional ratio (conER:conIR) is a general guideline for overhead athletes but is not specifically for swimmers [22]. Additionally, the lower (0.68) and upper (0.86) limits of the ideal functional ratio (eccER:conIR) were established using significantly lower angular velocities than those typically encountered during swimming techniques. The lower limit was determined by Drigny et al. [9], who monitored eighteen competitive swimmers over a season using an isokinetic dynamometer at 60°/s. Bak and Magnusson [22] defined the upper limit by evaluating the eccER:conIR in asymptomatic Danish competitive swimmers using an isokinetic dynamometer at 30°/s. These limitations may explain why our T0 assessments of the conventional and functional ratios did not align perfectly with the optimal reference values for injury prevention.

The effects of a weight program compared to an elastic band program were not considered significantly different. A systematic review conducted in 2019 concluded that training with elastic bands promotes strength gains comparable to those from weight resistance training across different populations and using various protocols [52]. In another study involving swimmers, it was observed that there is great variability in the electromyographic activity values of the shoulder muscles from exercise to exercise when executed with different instruments [35]. When analyzing the rotator cuff balance, the weight intervention showed a more positive impact on the conventional concentric ratio, while the elastic band protocol was more effective on the functional ratio. This information may assist in determining which preventive program is most appropriate for each swimmer. Generally, both instruments had good adherence; however, swimmers reported the portability of the elastic band as a positive feature.

After 12 weeks of experimental prevention programs, there has been an observed improvement in rotator cuff strength and balance. This duration aligns with the American College of Sports Medicine guidelines, which suggest a minimum of 10 weeks for achieving visible effects in strength and endurance programs [36]. Interestingly, studies [32,33] that reported no effects from preventive programs on strength and endurance variables in swimmers had a follow-up period of only 6 weeks, which may explain the absence of results. Similarly, another study [34] conducted in a different overhead sport found no benefits from an 8-week prevention program for shoulder injuries concerning strength and endurance variables.

It is important to note that there were no dropouts in any of the experimental groups. This contrasts with findings from similar studies [32,34]. The individual monitoring and correction provided to each swimmer likely contributed positively to this high adherence over the 12 weeks of intervention.

Implementing a prevention program for swimmer’s shoulder that includes five open kinetic chain exercises performed with weights or elastic bands can help minimize the natural rotator cuff imbalances during the competitive swimming season. This program should be carried out twice a week for 12 weeks, ideally with the assistance of a physiotherapist for individual monitoring and progression. An intensity level of 75% of 1RM has proven to be effective. It is advisable to introduce this preventive program during the pre-season, allowing athletes time to establish routines for the remainder of the competitive season. By adopting these preventive measures, swimmers can reduce the impact of an important risk factor for the most common injury in swimming, decreasing the likelihood of its occurrence.

One limitation of this trial is the small sample size within each group. Additionally, the fact that the research was conducted only with athletes from two swimming teams may impact its external validity. These factors could restrict the generalization of the results obtained. Furthermore, the increase in fatigue caused by the accumulation of training and competitions throughout the season also influenced the strength assessments.

It is essential to conduct similar investigations involving a larger number of competitive swimmers and swimming teams to better understand the effects of these preventive measures. Additionally, it is important to evaluate the impact of such interventions in other sports, particularly cyclical sports that often experience overuse injuries. Moreover, it is necessary to clarify the non-injury ranges of muscular balance across various sports, while also considering the high velocities involved in sports movements. This information will assist sports clinicians in identifying athletes at high risk of injury.

## 5. Conclusions

The implementation of two 12-week preventive programs for swimmers’ shoulder injuries did not result in an increase in IR and ER strength of the dominant shoulder at different tested velocities. However, the absence of a preventive program in the control group led to a reduction in strength, particularly in shoulder IRs. These preventive programs help minimize the normal rotator cuff imbalance developed during the competitive season for swimmers. No significant differences were observed in the effects of the two experimental programs, whether using weights or elastic bands. When selecting the appropriate equipment for these programs, it is essential to consider the swimmer’s previous clinical assessment and individual preferences. A 12-week duration with gradual load progression is adequate to assess some effects of these preventive programs. Individual monitoring is essential for improving swimmers’ adherence to this type of intervention.

## Figures and Tables

**Figure 1 healthcare-13-00538-f001:**
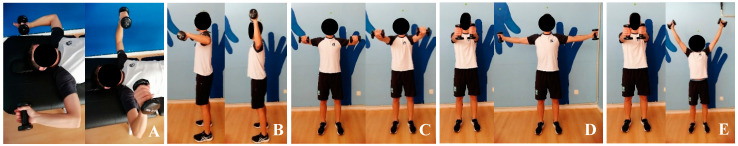
Illustration of the five exercises included in the weight program: (**A**) IR at 90°, (**B**) ER at 90°, (**C**) Scapular punches, (**D**) T’s, and (**E**) Y’s.

**Figure 2 healthcare-13-00538-f002:**
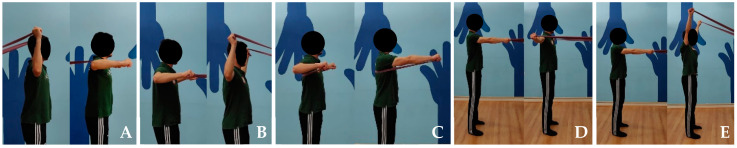
Illustration of the five exercises included in the elastic band program: (**A**) IR at 90°, (**B**) ER at 90°, (**C**) Scapular punches, (**D**) T’s, and (**E**) Y’s.

**Figure 3 healthcare-13-00538-f003:**
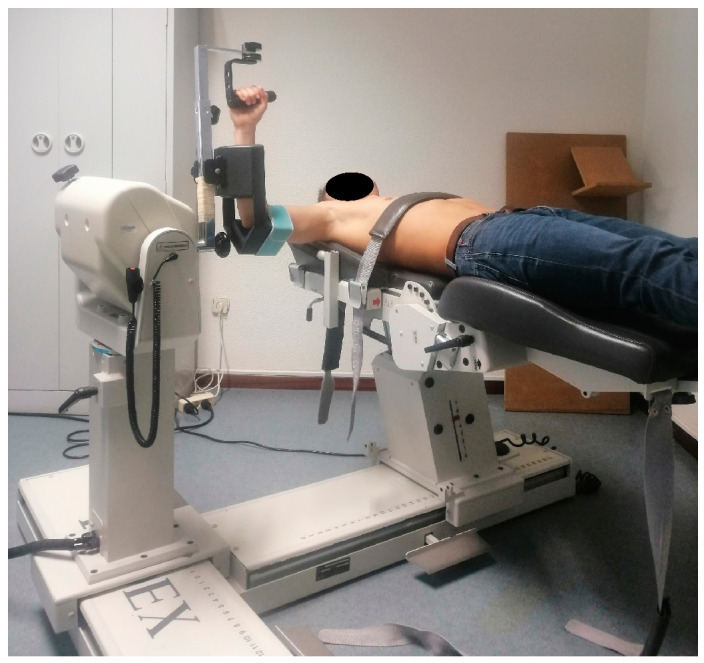
Subject position and stabilizing during the isokinetic dynamometer strength assessment.

**Figure 4 healthcare-13-00538-f004:**
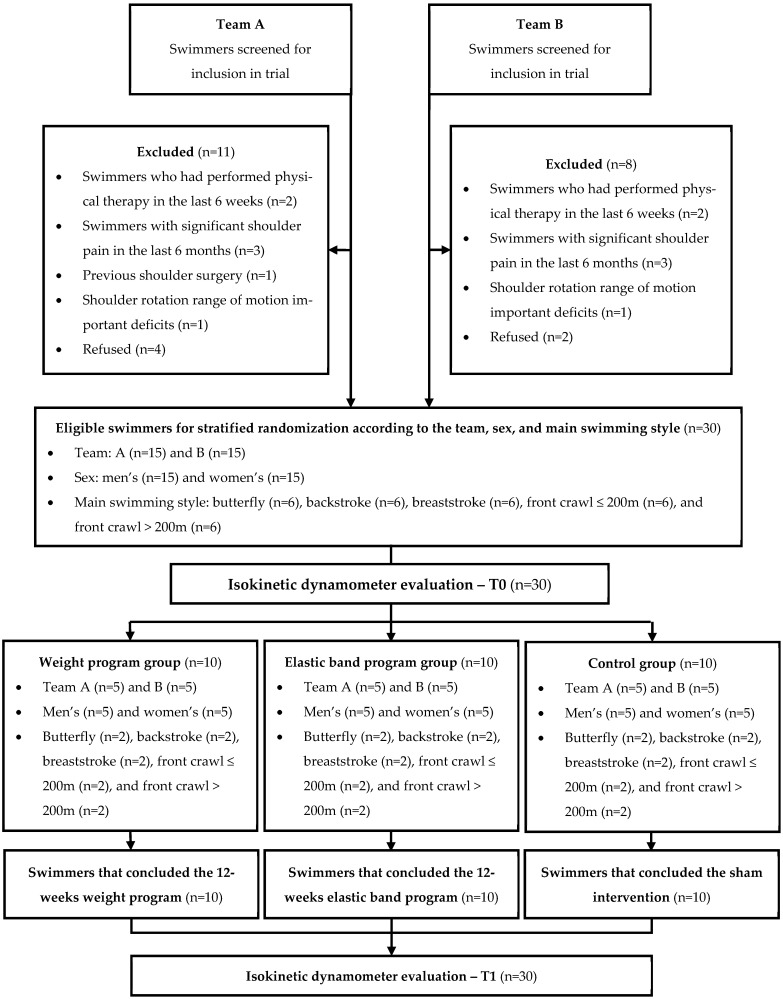
The flow of participants through the trial.

**Figure 5 healthcare-13-00538-f005:**
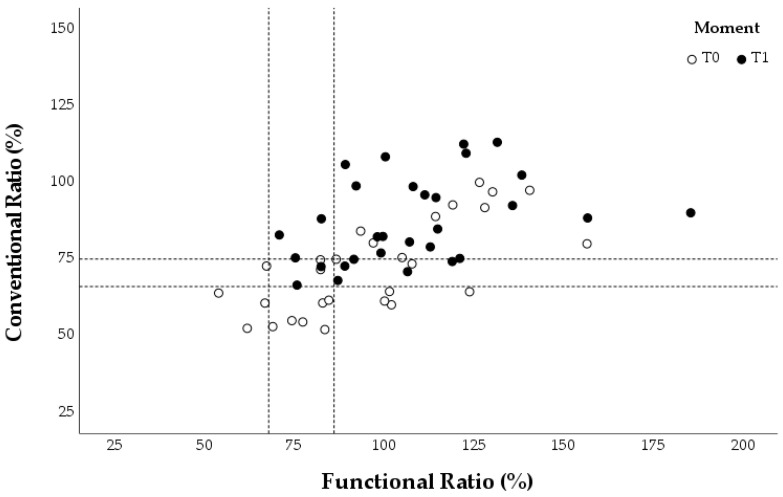
Conventional and functional ratios at 60°/s, 120°/s, and 180°/s for each swimmer in the weight program group. The reference values for preventing shoulder injuries are indicated with a dashed line.

**Figure 6 healthcare-13-00538-f006:**
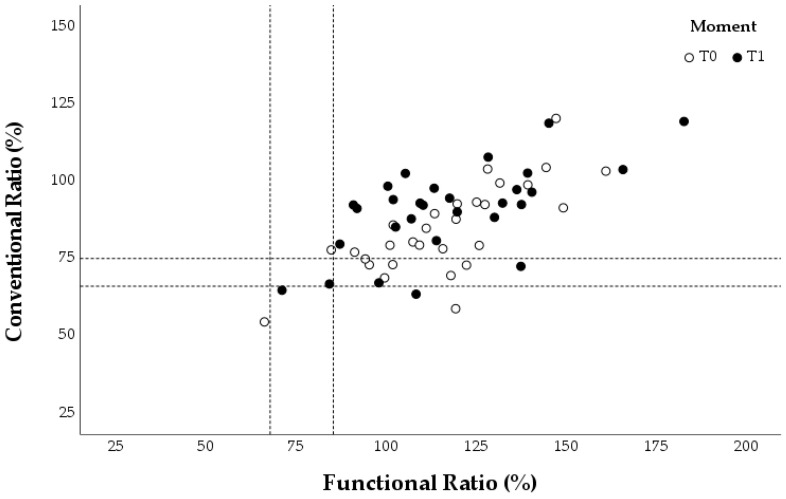
Conventional and functional ratios at 60°/s, 120°/s, and 180°/s for each swimmer in the elastic band program group. The reference values for preventing shoulder injuries are indicated with a dashed line.

**Figure 7 healthcare-13-00538-f007:**
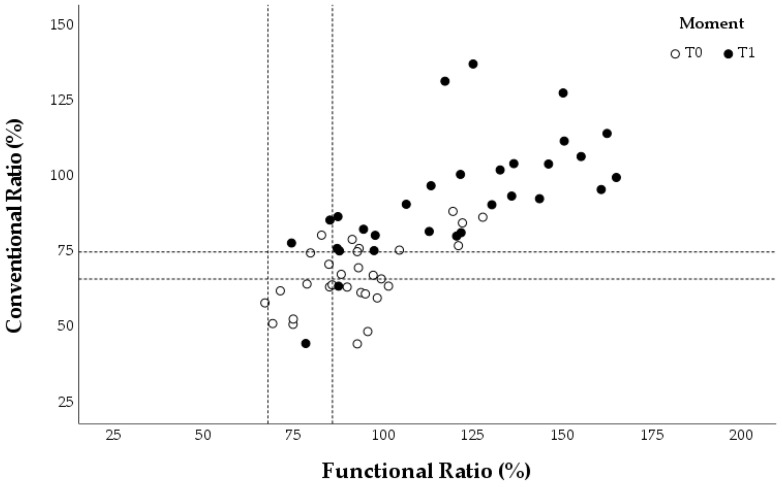
Conventional and functional ratios at 60°/s, 120°/s, and 180°/s for each swimmer in the control group. The reference values for preventing shoulder injuries are indicated with a dashed line.

**Table 1 healthcare-13-00538-t001:** Description of five exercises that constitute the weight and elastic band programs.

Exercise	Weight Program	Elastic Band Program
IR at 90°	Started in a supine position, 90° shoulder ABD, 90° ER, and 90° elbow FLX. Movement: 90° shoulder IR and return to the starting position (5 s).	Started in a SP, 90° shoulder ABD, 90° elbow FLX, and hands at the same height as shoulders. The elastic band should be fixed at shoulder height. Movement: 90° shoulder IR (5 s) and return to the starting position (5 s).
ER at 90°	Started in a SP, 90° shoulder ABD, 90° elbow FLX, and hands at the same height as shoulders. Movement: 90° shoulder ER (5 s) and return to the starting position (5 s).	Started in a SP, 90° shoulder ABD, 90 elbow FLX, and hands at the same height as shoulders. The elastic band should be fixed at shoulder height. Movement: 90° shoulder ER (5 s) and return to the starting position (5 s).
Scapular punches	Started in a SP, 90° shoulder ABD, 90° elbow FLX, and hands at the same height as shoulders. Movement: Elbow EXT (5 s) and return to the starting position (5 s).	Started in a SP, 90° shoulder ABD, 90° elbow FLX, and hands at the same height as shoulders. The elastic band should be fixed at shoulder height. Movement: Elbow EXT (5 s) and return to the starting position (5 s).
T’s	Started in a SP, 90° shoulder FLX, and maximum elbow EXT. Movement: 90° shoulder HABD (5 s) and return to the starting position (5 s).	Started in a SP, 90° shoulder FLX, and maximum elbow EXT. The elastic band should be fixed at shoulder height. Movement: 90° shoulder HABD (5 s) and return to the starting position (5 s).
Y’s	Started in a SP, 90° shoulder FLX, and maximum elbow EXT. Movement: 90° shoulder HABD and maximum shoulder FLX (5 s) and return to the starting position (5 s).	Started in a SP, 90° shoulder FLX, and maximum elbow EXT. The elastic band should be fixed at shoulder height. Movement: 90° shoulder HABD and maximum shoulder FLX (5 s) and return to the starting position (5 s).

SP—standing position; s—seconds; ABD—abduction; ER—external rotation; IR—internal rotation; FLX—flexion; EXT—extension; HABD—horizontal abduction.

**Table 2 healthcare-13-00538-t002:** Sample characterization and respective homogeneity of variance between the three groups.

Variables	WP Mean ± SD	EBP Mean ± SD	Control Mean ± SD	*p*-Value
Age (years)	19.90 ± 2.92	19.60 ± 2.99	19.00 ± 3.62	0.506
Body Mass (kg)	68.86 ± 8.49	68.80 ± 14.76	65.57 ± 11.77	0.879
Height (m)	1.72 ± 0.08	1.70 ± 0.11	1.73 ± 0.09	0.774
Body Mass Index (kg/m^2^)	23.19 ± 2.24	23.64 ± 3.34	21.79 ± 1.91	0.262
Fat mass (%)	17.02 ± 10.30	17.37 ± 8.65	15.40 ± 7.19	0.700
Lean mass (kg)	55.23 ± 10.60	57.66 ± 12.83	52.04 ± 13.66	0.868
Bone mass (kg)	2.91 ± 0.51	3.03 ± 0.64	2.75 ± 0.69	0.706
Swimming practice (years)	12.10 ± 4.18	9.40 ± 4.81	12.30 ± 4.17	0.274
Competitive swimming (years)	8.90 ± 3.28	6.10 ± 3.73	7.90 ± 3.38	0.206
Weekly swimming training (hours)	16.80 ± 5.01	14.20 ± 6.29	16.60 ± 6.87	0.577

SD—standard deviation; WP—weight program; EBP—elastic band program.

**Table 3 healthcare-13-00538-t003:** The homogeneity of variance between the 3 sample groups for the PT at T0.

		WP PT (Nm/kg)	EBP PT (Nm/kg)	Control PT (Nm/kg)	*p*-Value
60°/s	conIR	36.06 ± 9.71	33.17 ± 9.25	37.99 ± 13.49	0.619
	eccIR	37.30 ± 7.74	38.62 ± 14.47	41.67 ± 19.08	0.790
	conER	24.98 ± 6.63	27.62 ± 7.95	25.42 ± 10.25	0.756
	eccER	31.69 ± 7.11	36.14 ± 11.61	34.51 ± 13.58	0.667
120°/s	conIR	30.84 ± 9.22	32.13 ± 11.56	34.66 ± 13.53	0.756
	eccIR	36.71 ± 8.05	36.43 ± 11.96	38.01 ± 17.24	0.959
	conER	23.95 ± 6.91	26.90 ± 8.88	23.63 ± 11.58	0.691
	eccER	29.74 ± 7.46	35.47 ± 11.95	30.79 ± 12.80	0.473
180°/s	conIR	28.63 ± 9.70	28.22 ± 10.73	32.07 ± 13.90	0.719
	eccIR	38.11 ± 6.99	39.81 ± 12.95	39.53 ± 19.00	0.958
	conER	21.33 ± 5.07	24.26 ± 9.45	22.65 ± 11.64	0.774
	eccER	31.08 ± 7.74	37.28 ± 12.10	31.26 ± 14.64	0.464

WP—weight program; EBP—elastic band program; PT—peak torque; con—concentric; ecc—eccentric; IR—internal rotation; ER—external rotation.

**Table 4 healthcare-13-00538-t004:** Intra-group analyses in PT between T0 and T1.

		T0 PT (Nm/kg)	T1 PT (Nm/kg)	Difference PT (Nm/kg)	*p*-Value
Weight Program	conIR at 60°/s	36.06 ± 9.71	30.41 ± 11.93	−5.65	0.047 *
	eccIR at 60°/s	37.30 ± 7.74	37.16 ± 10.80	−0.14	0.959
	conER at 60°/s	24.98 ± 6.63	25.96 ± 8.45	0.98	0.506
	eccER at 60°/s	31.69 ± 7.11	29.84 ± 7.78	−1.85	0.315
	conIR at 120°/s	30.84 ± 9.22	29.57 ± 10.77	−1.27	0.537
	eccIR at 120°/s	36.71 ± 8.05	36.91 ± 10.74	0.20	1.000
	conER at 120°/s	23.95 ± 6.91	24.41 ± 7.05	0.46	0.705
	eccER at 120°/s	29.74 ± 7.46	29.03 ± 7.54	−0.71	0.704
	conIR at 180°/s	28.63 ± 9.70	26.21 ± 9.77	−2.42	0.215
	eccIR at 180°/s	38.11 ± 6.99	37.49 ± 9.65	−0.62	0.757
	conER at 180°/s	21.33 ± 5.07	22.74 ± 7.01	1.41	0.294
	eccER at 180°/s	31.08 ± 7.74	30.02 ± 8.83	−1.06	0.642
Elastic Band Program	conIR at 60°/s	33.17 ± 9.25	32.65 ± 13.57	−0.52	0.445
	eccIR at 60°/s	38.62 ± 14.47	38.08 ± 19.37	−0.54	0.575
	conER at 60°/s	27.62 ± 7.95	28.42 ± 8.12	0.80	0.595
	eccER at 60°/s	36.14 ± 11.61	35.57 ± 13.38	−0.57	0.818
	conIR at 120°/s	32.13 ± 11.56	30.95 ± 15.44	−1.18	0.074
	eccIR at 120°/s	36.43 ± 11.96	36.63 ± 18.98	0.20	0.594
	conER at 120°/s	26.90 ± 8.88	26.84 ± 9.12	−0.06	0.977
	eccER at 120°/s	35.47 ± 11.95	33.03 ± 11.15	−2.44	0.386
	conIR at 180°/s	28.22 ± 10.73	27.37 ± 13.81	−0.85	0.648
	eccIR at 180°/s	39.81 ± 12.95	37.48 ± 21.51	−2.33	0.386
	conER at 180°/s	24.26 ± 9.45	24.74 ± 10.70	0.48	0.828
	eccER at 180°/s	37.28 ± 12.10	35.36 ± 13.47	−1.92	0.301
Control	conIR at 60°/s	37.99 ± 13.49	28.15 ± 10.65	−9.84	0.002 *
	eccIR at 60°/s	41.67 ± 19.08	37.03 ± 16.37	−4.64	0.036 *
	conER at 60°/s	25.42 ± 10.25	25.85 ± 9.38	0.43	0.788
	eccER at 60°/s	34.51 ± 13.58	30.54 ± 12.50	−3.97	0.048 *
	conIR at 120°/s	34.66 ± 13.53	26.07 ± 10.35	−8.59	0.005 *
	eccIR at 120°/s	38.01 ± 17.24	37.08 ± 16.05	−0.93	0.627
	conER at 120°/s	23.63 ± 11.58	24.02 ± 9.68	0.39	0.843
	eccER at 120°/s	30.79 ± 12.80	30.18 ± 12.88	−0.61	0.675
	conIR at 180°/s	32.07 ± 13.90	24.53 ± 10.46	−7.54	0.002 *
	eccIR at 180°/s	39.53 ± 19.00	37.15 ± 16.10	−2.38	0.264
	conER at 180°/s	22.65 ± 11.64	22.20 ± 9.70	−0.45	0.771
	eccER at 180°/s	31.26 ± 14.64	31.61 ± 13.71	0.35	0.680

* Statistically significant difference. PT—peak torque; con—concentric; ecc—eccentric; IR—internal rotation; ER—external rotation.

## Data Availability

Data are contained within the article and Appendix A.

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
