# Peer review of "Effect of Preventive Exercise Programs for Swimmer’s Shoulder Injury on Rotator Cuff Torque and Balance in Competitive Swimmers: A Randomized Controlled Trial"

_healthcare, 2025, doi:10.3390/healthcare13050538_

Round 1

Reviewer 1 Report

Comments and Suggestions for Authors

Introduction has a clear presentation of the problem and its prevalence, it is well-supported with relevant literature,  and has clear research objectives and hypothesis.However, there are some claims could benefit from more recent citations (e.g., references from 1997). The gap in literature could be more explicitly stated.

Methodology has detailed description of procedures. However, sample size calculation should include more details about effect size selection, control group activities could be better described. Even the size effect is ok, the sample is too small for generalization.

Results are well presented, but discusion is a limited discussion of clinical significance, and could better address potential mechanisms, with more emphasis needed on practical applications. Limited discussion of unexpected findings and limited generalization should be explained.

About APA style, some inconsistencies in citation format and reference list needs formatting corrections, like some in-text citations missing page numbers for direct quotes. Review APA.

Author Response

Thank you very much for the revisions. The article was corrected, and we tried to respond to all the points mentioned.

Comment 1: Introduction has a clear presentation of the problem and its prevalence, it is well-supported with relevant literature, and has clear research objectives and hypothesis. However, there are some claims could benefit from more recent citations (e.g., references from 1997). The gap in literature could be more explicitly stated.

Response 1: We decided to include the reference from 1997 because it is a reference article that identifies the upper limit of the non-injury functional ratio range for swimmer's shoulder injury. Despite its publication date, this reference remains a cornerstone in the field, providing foundational insights into rotator cuff mechanics and injury prevention that continue to inform contemporary research on swimmer’s shoulder.

Comment 2: Methodology has detailed description of procedures. However, sample size calculation should include more details about effect size selection, control group activities could be better described. Even the size effect is ok, the sample is too small for generalization.

Response 2: We include more details about effect size selection (lines 248-261), and control group activities (lines 189-191).

“The sample size calculation was performed using G*Power software (Franz Faul, Edgar Erdfelder, Axel Buchner, Universität Kiel, Kiel, Germany, version 3.1.9.4). The statistical test selected was designed to compare mean differences between independent groups. The effect size used for this calculation was derived from a similar clinical trial [24], which involved 56 participants divided into three groups: an experimental group of 20 swimmers who participated in both a preventive strength program and swimming training, a training group of 20 swimmers who only engaged in swimming training, and a control group consisting of 16 active non-swimmers. Based on this trial [24], an effect size (Cohen’s d) of 2.81 was identified, representing the smallest difference considered statistically significant between the experimental and training groups when assessing IR and ER PT at 60º/s and 180º/s of the swimmers' dominant shoulder. Using an alpha level of 0.01, a power of 0.99, and an allocation ratio of 1:1, the software projected a minimum total sample size of 24 swimmers (8 per group). To account for potential dropouts, a final sample size of 30 swimmers (10 per group) was considered.”

“To ensure ethical equity among the sample groups and considering the potential benefits of the exercise programs tested, the control group was guaranteed access to either the weight or elastic band program after the conclusion of the trial.”

Comment 3: Results are well presented, but discusion is a limited discussion of clinical significance, and could better address potential mechanisms, with more emphasis needed on practical applications. Limited discussion of unexpected findings and limited generalization should be explained.

Response 3: The discussion has been extensively improved. A paragraph has been added to discuss the practical applications of the findings (lines 569-578), along with another paragraph that outlines the main limitations (lines 579-583).

“Implementing a prevention program for swimmers' shoulder that includes five open kinetic chain exercises performed with weights or elastic bands can help minimize the natural rotator cuff imbalances during the competitive swimming season. This program should be carried out twice a week for 12 weeks, ideally with the assistance of a physiotherapist for individual monitoring and progression. An intensity level of 75% of 1RM has proven to be effective. It is advisable to introduce this preventive program during the pre-season, allowing athletes time to establish routines for the remainder of the competitive season. By adopting these preventive measures, swimmers can reduce the impact of an important risk factor for the most common injury in swimming, decreasing the likelihood of its occurrence.”

“One limitation of this trial is the small sample size within each group. Additionally, the fact that the research was conducted only with athletes from two swimming teams may impact its external validity. These factors could restrict the generalization of the results obtained. Furthermore, the increase in fatigue caused by the accumulation of training and competitions throughout the season also influenced the strength assessments.”

Comment 4: About APA style, some inconsistencies in citation format and reference list needs formatting corrections, like some in-text citations missing page numbers for direct quotes. Review APA.

Response 4: All bibliographical references have been reviewed and corrected as needed, considering the journal's citation format requirements.

Reviewer 2 Report

Comments and Suggestions for Authors

Dear authors,

First of all congratulation on submitting your paper for publication and thank you for the invitation to review your study “Effect of Preventive Exercise Programs for Swimmer’s Shoulder Injury on Rotator Cuff Torque and Balance in Competitive   Swimmers: A Randomized Controlled Trial ”. The study introduced an interesting topic the Swimmer's shoulder that could be of interest for a broad professionals involved in sport preventions injuries and recovery. 

Please find some specific comments below.

General: there are some abbreviations that make the text difficult to read, please use the full name in the first paragraph of every section (introduction, method, results and discussion), then the abbreviation.

Introduction

  • Swimmer's shoulder is the most prevalent injury in swimmers, it results not only from the excessive impingement of the rotator cuff under the coracoacromial arch, but also from fatigue. The activation of several muscles lead to fatigue after swim training. Fatigue may affect the shoulder’s strength, proprioception, and range of motion, representing possible risk factors for overuse shoulder injury; decreased muscle endurance or either hypo- or hyperactivation of shoulder muscles could lead to abnormal movement of the glenohumeral and scapulothoracic joints. Please add also fatigue as factor and take in to consideration the following articles:

Buoite Stella A, Cargnel A, Raffini A, Mazzari L, Martini M, Ajčević M, Accardo A, Deodato M, Murena L. Shoulder Tensiomyography and Isometric Strength in Swimmers Before and After a Fatiguing Protocol. J Athl Train. 2024 Jul 1;59(7):738-744. doi: 10.4085/1062-6050-0265.23. PMID: 38014804; PMCID: PMC11277270.

Yoma M, Herrington L, Mackenzie TA, Almond TA. Training Intensity and Shoulder Musculoskeletal Physical Quality Responses in Competitive Swimmers. J Athl Train. 2021 Jan 1;56(1):54-63. doi: 10.4085/1062-6050-0357.19. PMID: 33176360; PMCID: PMC7863595.

  • I would first explain the hypothesis of the study and then the aims of the study
  • Could you split the aim of the study in a primary and secondary aims?

METHOD

  • Details of trial registration number should be included
  • Details on the blinding process are limited. . Expanding on how blinding was maintained for participants and assessors could help minimize bias and clarify the study's validity, especially for sham group.
  • The manuscript does not clearly justify the chosen sample size or discuss how it ensures sufficient power to detect expected results.
  • The study involved both male and female subjects, that could be an important variable, please add information regard in which phase of menstrual cycle were the female subjects assessed.

Results

  • Tables presented have so many lines, please could you kindly use only three main lines? Especially for tables 2

Discussion

  • This section should be whole revise, I try to suggest some way
  • Please avoid repeating in the discussion section results such as p-value or see table 1 or figure 2…it is not a repetition of results section.
  • I suggest a structure more congruent in flow of ideas and continuity of thought: first paragraph only your main findings; second paragraph comparing your result with previous studies; third add a more comprehensive discussion of the study's limitations, including the small sample size; fourth add a personal general hypothesis for your findings; finally it could be interesting suggesting future work.

Author Response

Dear authors,

First of all congratulation on submitting your paper for publication and thank you for the invitation to review your study “Effect of Preventive Exercise Programs for Swimmer’s Shoulder Injury on Rotator Cuff Torque and Balance in Competitive   Swimmers: A Randomized Controlled Trial ”. The study introduced an interesting topic the Swimmer's shoulder that could be of interest for a broad professionals involved in sport preventions injuries and recovery. 

Thank you very much for the revisions. The article was corrected, and we tried to respond to all the points mentioned.

Please find some specific comments below.

Comment 1: General: there are some abbreviations that make the text difficult to read, please use the full name in the first paragraph of every section (introduction, method, results and discussion), then the abbreviation.

Response 1: The full name of the abbreviations was added in the first paragraph of every section (lines 48, 52-55, 122, 138, 194, 321, 409-410, 489-490).

Introduction

Comment 2: Swimmer's shoulder is the most prevalent injury in swimmers, it results not only from the excessive impingement of the rotator cuff under the coracoacromial arch, but also from fatigue. The activation of several muscles lead to fatigue after swim training. Fatigue may affect the shoulder’s strength, proprioception, and range of motion, representing possible risk factors for overuse shoulder injury; decreased muscle endurance or either hypo- or hyperactivation of shoulder muscles could lead to abnormal movement of the glenohumeral and scapulothoracic joints. Please add also fatigue as factor and take in to consideration the following articles:

Buoite Stella A, Cargnel A, Raffini A, Mazzari L, Martini M, Ajčević M, Accardo A, Deodato M, Murena L. Shoulder Tensiomyography and Isometric Strength in Swimmers Before and After a Fatiguing Protocol. J Athl Train. 2024 Jul 1;59(7):738-744. doi: 10.4085/1062-6050-0265.23. PMID: 38014804; PMCID: PMC11277270.

Yoma M, Herrington L, Mackenzie TA, Almond TA. Training Intensity and Shoulder Musculoskeletal Physical Quality Responses in Competitive Swimmers. J Athl Train. 2021 Jan 1;56(1):54-63. doi: 10.4085/1062-6050-0357.19. PMID: 33176360; PMCID: PMC7863595.

Response 2: Both articles were included, along with adding fatigue as a risk factor (line 40-41).

“Several risk factors contribute to this painful syndrome, including … fatigue [10,11,13]…”

Comment 3: I would first explain the hypothesis of the study and then the aims of the study.

Response 3: We maintained the objectives before the hypothesis to transition from a broader perspective to a more specific one. Usually in a research paper, the study aim should come first, followed by the hypothesis because the study aim provides a broad statement about what the research intends to achieve. It sets the context and highlights the purpose of the study. The hypothesis is a more specific statement predicting the expected outcome of the study. It is usually derived from the study aim and existing literature.

Comment 4: Could you split the aim of the study in a primary and secondary aims?

Response 4: The primary and secondary aims were included (97-104).

“The primary aim is to verify the effectiveness of two 12-week preventive programs for swimmers’ shoulder injuries on the IR and ER PT and respective conventional and functional ratios. These programs included regular monitoring and progression over time. One program utilized weights while the other employed elastic bands. The secondary aims include examining the differences between the weights and elastic bands programs, evaluating whether a 12-week program duration leads to significant results, and analyzing the impact of individualized monitoring on reducing dropout rates during the follow-up period.”

METHOD

Comment 5: Details of trial registration number should be included

Response 5: The trial registration number was included (113-114).

“The protocol for the randomized controlled trial has been registered on ClinicalTrials.gov with the registration number NCT06552585.”

Comment 6: Details on the blinding process are limited. . Expanding on how blinding was maintained for participants and assessors could help minimize bias and clarify the study's validity, especially for sham group.

Response 6: Additional information regarding blinding and the study's validity were included (lines 271-274, 279-280, 285-288).

“The allocation process was concealed within each stratum, employing a computer-generator random number managed by a member of the research team. The participants were unaware of the existence of other sample groups.”

“…supervised by a sports physiotherapist who was external to the study and unaware of the previous procedures or the study's objectives.”

“All in-person and online sessions were individualized to ensure that each athlete remained unaware of the other groups. The sports physiotherapist … did not have contact with those in the control group.”

Comment 7: The manuscript does not clearly justify the chosen sample size or discuss how it ensures sufficient power to detect expected results.

Response 7: The process for calculating the sample size was explained in more detail (lines 248-261).

“The sample size calculation was performed using G*Power software (Franz Faul, Edgar Erdfelder, Axel Buchner, Universität Kiel, Kiel, Germany, version 3.1.9.4). The statistical test selected was designed to compare mean differences between independent groups. The effect size used for this calculation was derived from a similar clinical trial [24], which involved 56 participants divided into three groups: an experimental group of 20 swimmers who participated in both a preventive strength program and swimming training, a training group of 20 swimmers who only engaged in swimming training, and a control group consisting of 16 active non-swimmers. Based on this trial [24], an effect size (Cohen’s d) of 2.81 was identified, representing the smallest difference considered statistically significant between the experimental and training groups when assessing IR and ER PT at 60º/s and 180º/s of the swimmers' dominant shoulder. Using an alpha level of 0.01, a power of 0.99, and an allocation ratio of 1:1, the software projected a minimum total sample size of 24 swimmers (8 per group). To account for potential dropouts, a final sample size of 30 swimmers (10 per group) was considered.”

Comment 8: The study involved both male and female subjects, that could be an important variable, please add information regard in which phase of menstrual cycle were the female subjects assessed.

Response 8: The information was added (lines 129-130).

“The assessment of female athletes was conducted outside of their menstruation period.”

Results

Comment 9: Tables presented have so many lines, please could you kindly use only three main lines? Especially for tables 2

Response 9: Only three main lines were included in Table 2.

Discussion

This section should be whole revise, I try to suggest some way

Comment 10: Please avoid repeating in the discussion section results such as p-value or see table 1 or figure 2…it is not a repetition of results section.

Response 10: All references to p-values, figures, or tables have been removed from the Discussion section.

Comment 11: I suggest a structure more congruent in flow of ideas and continuity of thought: first paragraph only your main findings; second paragraph comparing your result with previous studies; third add a more comprehensive discussion of the study's limitations, including the small sample size; fourth add a personal general hypothesis for your findings; finally it could be interesting suggesting future work.

Response 11: The discussion has been extensively improved. We tried to follow the suggested discussion model:

  • Main findings: 1st and 2nd paragraphs (488-502)
  • Comparison of the results with previous studies: 3rd, 4th, 5th, 6th, 7th and 8th paragraphs (503-568)
  • Personal general hypothesis and practical applications of the findings: 9th paragraph (569-578)
  • Study's limitations: 10th paragraph (579-583)
  • Suggestions for future works: 11th paragraph (584-590)

Reviewer 3 Report

Comments and Suggestions for Authors

Introduction:

  1. Injury Mechanism: The link between rotator cuff imbalances and shoulder injuries could be better explained beyond just impingement to strengthen the study's rationale.
  2. References: Some claims, like the effectiveness of strength prevention programs, would benefit from more supporting references to increase credibility.
  3. Study Rationale: The comparison between weights and elastic bands is interesting. However, it would be helpful to explain why this distinction matters for injury prevention in swimmers.
  4. Literature Gaps: The mention of mixed results from previous studies should be expanded with a brief explanation of the factors contributing to these inconsistencies.
  5. Hypothesis Support: Clarifying why a 12-week intervention is expected to be effective based on previous research would enhance the introduction.

Methods Section:

  1. Randomization Process: Additional details on how randomization was performed (e.g., allocation concealment, stratification) would improve transparency.
  2. Blinding: Clarify how blinding was ensured for both participants and care providers to reduce potential bias.
  3. Recruitment Process: Include more information on how participants were approached, how many were initially contacted, and the number of declines.
  4. Exercise Supervision: Explain the methods of online supervision via WhatsApp/Zoom and how adherence was monitored.
  5. 1RM Measurement: Specify the protocol for 1RM assessment, including warm-up procedures and reliability measures.
  6. Sham Intervention: Discuss potential placebo effects and the impact of the control group’s lack of progressive loading on results.
  7. Isokinetic Testing Reliability: Indicate whether test-retest reliability was assessed and provide relevant coefficients if available.
  8. Participant Positioning: Justify the use of the supine position for isokinetic testing and discuss any limitations compared to the seated position.
  9. Sample Size and Dropout Rate: Provide details on the sample size calculation and expected dropout rates to ensure statistical power.

Results:

  1. Clarity of Results: The results section is well-structured, but a clearer summary of the key findings in the text could improve readability, especially for those less familiar with statistical data.
  2. Statistical Reporting: Ensure consistent reporting of p-values (e.g., three decimal places throughout). Some statistical results could be explained in the text rather than relying solely on tables.
  3. Grouping and Comparison: The stratified randomization method is well-explained, but providing a rationale for the chosen stratification factors (e.g., swimming style) would add clarity. Additionally, it would be helpful to highlight which intervention showed the most meaningful effects.
  4. Tables and Figures: While tables are detailed, summarizing the key takeaways in the text would improve accessibility. Referencing Figure 4 more explicitly, if available, would guide readers effectively.
  5. Additional Considerations: If there were no dropouts, comments on participant compliance or adherence would be helpful. Also, acknowledging potential study limitations, such as sample size or external validity, would enhance the discussion.

Discussion:

  1. Results Clarity: The discussion would benefit from a clearer, more concise summary of the most important findings, especially for readers who may not be familiar with statistical details.
  2. Contextualizing Decreases in PT: The decrease in concentric IR PT at 60°/s in the weight program group should be explained in greater detail to emphasize its significance and relevance to shoulder health in swimmers.
  3. Practical Implications: The practical application of the findings should be emphasized more. This could focus on how the experimental programs can help prevent injuries in real-world training settings.
  4. Linking to Previous Research: A deeper comparison of the results with similar studies would strengthen the interpretation and highlight whether observed trends align with or differ from previous research.
  5. Limitations: The discussion could address study limitations such as sample size or external validity, offering a more balanced perspective and clearer directions for future research.
  6. Further Research: It would be helpful to expand on the potential areas for future research. This could include how these exercise programs might affect other types of injuries or be applied in different sports contexts.

Author Response

Thank you very much for the revisions. The article was corrected, and we tried to respond to all the points mentioned.

Introduction:

Comment 1: Injury Mechanism: The link between rotator cuff imbalances and shoulder injuries could be better explained beyond just impingement to strengthen the study's rationale.

Response 1: The link between rotator cuff imbalances and shoulder injuries was better explained (lines 45-47).

“These muscle imbalances can lead to abnormal movements of the glenohumeral and scapulothoracic joints, which may continuously aggravate susceptible tissues [13].”

Comment 2: References: Some claims, like the effectiveness of strength prevention programs, would benefit from more supporting references to increase credibility.

Response 2: Two systematic reviews were added to the five clinical trials that reported the effectiveness of strength prevention programs on rotator cuff balance (line 73).

Comment 3: Study Rationale: The comparison between weights and elastic bands is interesting. However, it would be helpful to explain why this distinction matters for injury prevention in swimmers.

Response 3: This information was explained in more detail (lines 85-88).

“In shoulder exercises, the elastic band typically produces more electromyography activity in the posterior muscles, while weights engage the lateral and anterior muscles [33]. This significant distinction may lead to different potential effects from each exercise program.”

Comment 4: Literature Gaps: The mention of mixed results from previous studies should be expanded with a brief explanation of the factors contributing to these inconsistencies.

Response 4: This information was explained in more detail (lines 73-78).

“However, some authors report no effect of these preventive programs on strength and endurance variables in swimmers [30,31] and other overhead sports athletes [32]. In general, these studies are characterized by a big heterogeneity: in the quantity and type of exercises performed during the program, in the kind of instrument used in the exercises, and in the monitoring, duration, and progression of the program over time.”

Comment 5: Hypothesis Support: Clarifying why a 12-week intervention is expected to be effective based on previous research would enhance the introduction.

Response 5: The rationale for selecting a 12-week duration for the interventions was explained more clearly (lines 91-93).

“According to American College of Sports Medicine guidelines, strength or endurance exercise programs demonstrated significant improvements only after 10 weeks of training”

Methods Section:

Comment 6: Randomization Process: Additional details on how randomization was performed (e.g., allocation concealment, stratification) would improve transparency.

Response 6: The information about the randomization process was improved (lines 270-273).

“Participants were initially screened to determine their respective strata. The allocation process was concealed within each stratum, employing a computer-generator random number managed by a member of the research team. The participants were unaware of the existence of other sample groups.”

Comment 7: Blinding: Clarify how blinding was ensured for both participants and care providers to reduce potential bias.

Response 7: The information about blinding was clarified (lines 273-274 and lines 279-288).

“The participants were unaware of the existence of other sample groups.”

“…supervised by a sports physiotherapist who was external to the study and unaware of the previous procedures or the study's objectives.”

“All in-person and online sessions were individualized to ensure that each athlete remained unaware of the other groups. The sports physiotherapist … did not have contact with those in the control group.”

Comment 8: Recruitment Process: Include more information on how participants were approached, how many were initially contacted, and the number of declines.

Response 8: This information was added (lines 308-311).

“The initial contact with the competitive swimmers was made through a phone call, during which the eligibility criteria and individual consent for participation in the trial were screened. Out of 49 athletes contacted, 30 were eligible and deemed suitable to participate.”

Comment 9: Exercise Supervision: Explain the methods of online supervision via WhatsApp/Zoom and how adherence was monitored.

Response 9: The online supervision was explained in more detail (lines 157-158 and 285-288).

“… the remaining sessions were monitored online via WhatsApp Messenger or Zoom Video Communications, Inc. Each session was individualized and lasted about 30 minutes on average.”

“All in-person and online sessions were individualized to ensure that each athlete remained unaware of the other groups. The sports physiotherapist also tracked the number of sessions completed by each athlete and did not have contact with those in the control group.”

Comment 10: 1RM Measurement: Specify the protocol for 1RM assessment, including warm-up procedures and reliability measures.

Response 10: The 1RM assessment description was improved (lines 144-147).

“The protocol for the 1RM test followed the guidelines outlined in Baechle & Early [42]. The T's exercise was selected for this assessment. Before the 1RM test, a warm-up consisting of ten repetitions of the T's exercise was performed using a very light load, either with weights or an elastic band.”

Comment 11: Sham Intervention: Discuss potential placebo effects and the impact of the control group’s lack of progressive loading on results.

Response 11: The procedures for the control group were improved (lines 189-191).

“To ensure ethical equity among the sample groups and considering the potential benefits of the exercise programs tested, the control group was guaranteed access to either the weight or elastic band program after the conclusion of the trial.”

Comment 12: Isokinetic Testing Reliability: Indicate whether test-retest reliability was assessed and provide relevant coefficients if available. Participant Positioning: Justify the use of the supine position for isokinetic testing and discuss any limitations compared to the seated position.

Response 12: Reproducibility and reliability of the isokinetic testing performed in a supine position has been included, along with a justification for choosing this position (lines 202-209).

“The chosen evaluation position is similar to the body position of athletes during swimming. This position demonstrates better reproducibility and reliability for assessing IR, ER, and concentric conventional ratio compared to both seated assessment and lying evaluation with the shoulder at 45º of abduction [44,46]. Forthomme et al. [46] conducted strength assessments in this position at angular velocities of 60°/s and 240°/s showed reproducibility values of 11.8 and 10.5 for IR, 8.9 and 7.5 for ER, and 7.6 and 7.8 for the conventional concentric ratio. The reliability values were noted as 15.9 and 15.5 for IR, 9.6 and 6.6 for ER, and 0.15 and 0.13 for the conventional concentric ratio.”

Comment 13: Sample Size and Dropout Rate: Provide details on the sample size calculation and expected dropout rates to ensure statistical power.

Response 13: More detailed information regarding sample size and dropout rates was provided (lines 248-261).

“ The sample size calculation was performed using G*Power software (Franz Faul, Edgar Erdfelder, Axel Buchner, Universität Kiel, Kiel, Germany, version 3.1.9.4). The statistical test selected was designed to compare mean differences between independent groups. The effect size used for this calculation was derived from a similar clinical trial [24], which involved 56 participants divided into three groups: an experimental group of 20 swimmers who participated in both a preventive strength program and swimming training, a training group of 20 swimmers who only engaged in swimming training, and a control group consisting of 16 active non-swimmers. Based on this trial [24], an effect size (Cohen’s d) of 2.81 was identified, representing the smallest difference considered statistically significant between the experimental and training groups when assessing IR and ER PT at 60º/s and 180º/s of the swimmers' dominant shoulder. Using an alpha level of 0.01, a power of 0.99, and an allocation ratio of 1:1, the software projected a minimum total sample size of 24 swimmers (8 per group). To account for potential dropouts, a final sample size of 30 swimmers (10 per group) was considered.”

Results:

Comment 14: Clarity of Results: The results section is well-structured, but a clearer summary of the key findings in the text could improve readability, especially for those less familiar with statistical data.

Response 14: We summarized the key findings at the end of the Results section (lines 484-486).

“In summary, a more significant decrease in the shoulder internal rotation peak torque values in the control group increased the conventional and functional ratios, moving these values further away from the non-injury zone, compared to the experimental groups.”

Comment 15: Statistical Reporting: Ensure consistent reporting of p-values (e.g., three decimal places throughout). Some statistical results could be explained in the text rather than relying solely on tables.

Response 15: All significant values have been included in the text of the Results section. We always preserve three decimal places in p-values (line 393).

Comment 16: Grouping and Comparison: The stratified randomization method is well-explained, but providing a rationale for the chosen stratification factors (e.g., swimming style) would add clarity. Additionally, it would be helpful to highlight which intervention showed the most meaningful effects.

Response 16: This information was clarified (lines 314-319)

“The different types of swimming training according to the team or the main swimmer's style and the strength differences between males and females were two variables that could influence the results. To minimize its impact, each group maintained equal representation, consisting of five swimmers from team A and five from team B, five men's and five women's, and two swimmers from each swimming style – butterfly, backstroke, breaststroke, front crawl ≤ 200m, and front crawl > 200m (Figure 4).”

Comment 17: Tables and Figures: While tables are detailed, summarizing the key takeaways in the text would improve accessibility. Referencing Figure 4 more explicitly, if available, would guide readers effectively.

Response 17: Figure 4 was referenced in the text (line 319) and its position was changed.

Comment 18: Additional Considerations: If there were no dropouts, comments on participant compliance or adherence would be helpful. Also, acknowledging potential study limitations, such as sample size or external validity, would enhance the discussion.

Response 18: This information was added (lines 323-325).

“No participant dropouts were recorded during the 12 weeks of follow-up. All participants in the experimental interventions completed the 24 individual sessions of their prevention program over 12 weeks.”

Discussion:

Comment 19: Results Clarity: The discussion would benefit from a clearer, more concise summary of the most important findings, especially for readers who may not be familiar with statistical details.

Response 19: A more concise summary of the most important findings was provided in the first two paragraphs of the Discussion section (lines 488-502).

Comment 20: Contextualizing Decreases in PT: The decrease in concentric IR PT at 60°/s in the weight program group should be explained in greater detail to emphasize its significance and relevance to shoulder health in swimmers.

Response 20: We decided to focus on the reductions in peak torque observed in the control group in comparison to the experimental groups (lines 515-518). Although the weight program group showed a significant decrease, it was smaller compared to the control group during the conIR at 60°/s evaluation. Additionally, the differences in the other assessments were similar between the elastic bands and weights. It is possible that increased fatigue among participants in the weight group or a lower performance capacity during this specific evaluation (conIR at 60°/s) could explain these results.  However, given the fact that other results were similar, we should be cautious about concluding.

“Furthermore, it is important to highlight those five of the six significant decreases in shoulder PT occurred in the control group. This suggests that experimental procedures may help minimize the expected declines in rotator cuff PT throughout the swimming season.”

Comment 21: Practical Implications: The practical application of the findings should be emphasized more. This could focus on how the experimental programs can help prevent injuries in real-world training settings.

Response 21: We added a paragraph about practical implications in the Discussion section (lines 569-578).

“Implementing a prevention program for swimmers' shoulder that includes five open kinetic chain exercises performed with weights or elastic bands can help minimize the natural rotator cuff imbalances during the competitive swimming season. This program should be carried out twice a week for 12 weeks, ideally with the assistance of a physiotherapist for individual monitoring and progression. An intensity level of 75% of 1RM has proven to be effective. It is advisable to introduce this preventive program during the pre-season, allowing athletes time to establish routines for the remainder of the competitive season. By adopting these preventive measures, swimmers can reduce the impact of an important risk factor for the most common injury in swimming, decreasing the likelihood of its occurrence.”

Comment 22: Linking to Previous Research: A deeper comparison of the results with similar studies would strengthen the interpretation and highlight whether observed trends align with or differ from previous research.

Response 22: We try to make a deeper comparison of the results with similar studies in the 3rd, 4th,5th,6th,7th, and 8th paragraphs of the Discussion section (lines 503-568).

Comment 23: Limitations: The discussion could address study limitations such as sample size or external validity, offering a more balanced perspective and clearer directions for future research.

Response 23: We improved a paragraph about limitations (lines 579-583).

“One limitation of this trial is the small sample size within each group. Additionally, the fact that the research was conducted only with athletes from two swimming teams may impact its external validity. These factors could restrict the generalization of the results obtained. Furthermore, the increase in fatigue caused by the accumulation of training and competitions throughout the season also influenced the strength assessments.”

Comment 24: Further Research: It would be helpful to expand on the potential areas for future research. This could include how these exercise programs might affect other types of injuries or be applied in different sports contexts.

Response 24: The last paragraph of the Discussion section explains this topic (lines 584-590).

“It is essential to conduct similar investigations involving a larger number of competitive swimmers and swimming teams to better understand the effects of these preventive measures. Additionally, it is important to evaluate the impact of such interventions in other sports, particularly cyclical sports that often experience overuse injuries. Moreover, it is necessary to clarify the non-injury ranges of muscular balance across various sports, while also considering the high velocities involved in sports movements. This information will assist sports clinics in identifying athletes at high risk of injury.”

Round 2

Reviewer 2 Report

Comments and Suggestions for Authors

well done

Author Response

Comment 1: well done

Response 1: Thank you very much for your attention.

Reviewer 3 Report

Comments and Suggestions for Authors
  • Simplify Complex Sentences:
    Some sentences are long and complex, making them hard to follow. Try breaking them into shorter, clearer sentences to improve readability.
  • Enhance Flow Between Sections:
    The transition between topics like prevalence, risk factors, and strength ratios feels a bit abrupt. Adding smoother transitions will help the introduction flow more naturally.
  • Expand on Background Details:
    The introduction explains the injury well but could benefit from more details on how a swimmer’s shoulder develops and its impact on the swimmer’s performance. This would give readers a fuller understanding.
  • Update and Diversify References:
    While the references are solid, including more recent studies (from the past 5 years) would ensure the information is current. Adding sources on long-term rehab outcomes could also strengthen the background.
  • Clarify the Study Rationale:
    The aim is clear, but explicitly stating the research gap (e.g., the lack of consensus on the most effective preventive programs) would make the purpose of the study stronger.
  • Refine the Hypothesis:
    The hypothesis is good but could be more concise and direct. Simplifying the wording would make it easier for readers to understand the study’s focus.

Author Response

Thank you very much for the revisions. The article was corrected, and we tried to respond to all the points mentioned.

Introduction:

Comment 1: Simplify Complex Sentences: Some sentences are long and complex, making them hard to follow. Try breaking them into shorter, clearer sentences to improve readability.

Response 1: We have tried to simplify some sentences in the Introduction section. (lines 38-41, 65-71, 93-95)

“It is estimated that around 40-91% of swimmers will experience this issue at least once during their careers [2,6,9,10]. Furthermore, 20-35% of competitive swimmers experience a loss of training or competition time each year due to this injury [9,11].”

“Ellenbecker & Davies [24] concluded that to prevent shoulder injuries in overhead sports the conER:conIR ratio should be between 0.66 and 0.75. Consequently, the strength of the ER should be 2/3 compared to the IR of the shoulder. Regarding functional ratio, Bak & Magnusson [22] reported an eccER:conIR ratio = 0.86 in asymptomatic swimmers. Drigny et al. [9] considered that swimmers have a 4.5-fold increased risk of in-season shoulder injury with an eccER:conIR ratio < 0.68.”

“Furthermore, it is necessary to check how long the preventive program should be to produce an effect on the strength balance of the rotator cuff. Some studies indicated no effects of preventive programs used 8-week [36] or 6-week programs [32,33].”

Comment 2: Enhance Flow Between Sections: The transition between topics like prevalence, risk factors, and strength ratios feels a bit abrupt. Adding smoother transitions will help the introduction flow more naturally.

Response 2: Smoother transitions were added to the first part of the Introduction section. (lines 35-42, 46-47)

“Swimmer's shoulder is the most common injury among swimmers [1–6]. It is characterized by non-specific anterior shoulder pain resulting from the repetitive impingement of the rotator cuff under the coracoacromial arch [3,7]. This impingement can result in functional impairments and decreased athletic performance [8]. It is estimated that around 40-91% of swimmers will experience this issue at least once during their careers [2,6,9,10]. Furthermore, 20-35% of competitive swimmers experience a loss of training or competition time each year due to this injury [9,11]. The causes of a swimmer's shoulder are dynamic and multifactorial [5,7,12].”

“Modifiable risk factors are crucial for identifying athletes at increased injury risk. Many of these factors can be altered through therapeutic interventions [5].”

Comment 3: Expand on Background Details: The introduction explains the injury well but could benefit from more details on how a swimmer’s shoulder develops and its impact on the swimmer’s performance. This would give readers a fuller understanding.

Response 3: Information regarding the swimmer's shoulder development and its impact on performance was included in the Introduction section. (lines 35-38)

“It is characterized by non-specific anterior shoulder pain resulting from the repetitive impingement of the rotator cuff under the coracoacromial arch [3,7]. This impingement can result in functional impairments and decreased athletic performance [8].”

Comment 4: Update and Diversify References: While the references are solid, including more recent studies (from the past 5 years) would ensure the information is current. Adding sources on long-term rehab outcomes could also strengthen the background.

Response 4: We included two recent reviews in the Introduction section to ensure the information is current.

“McKenzie, A.; Larequi, S-A.; Hams, A.; Headrick, J.; Whiteley, R.; Duhig, S. Shoulder pain and injury risk factors in competitive swimmers: a systematic review. Scand J Med Sci Sports 2023, 33(12), 2396-2412. DOI: 10.1111/sms.14454.”

“Hill, L.; Mountjoy, M.; Miller, J. Non-shoulder injuries in swimming: a systematic review. Clin J Sport Med 2022, 32(3), 256-264. DOI: 10.1097/JSM.0000000000000903.”

Comment 5: Clarify the Study Rationale: The aim is clear, but explicitly stating the research gap (e.g., the lack of consensus on the most effective preventive programs) would make the purpose of the study stronger.

Response 5: The research gap was added to the primary aim. (lines 104-105)

“This objective seeks to clarify the lack of consensus on the most effective preventive programs.”

Comment 6: Refine the Hypothesis: The hypothesis is good but could be more concise and direct. Simplifying the wording would make it easier for readers to understand the study’s focus.

Response 6: We have simplified the wording of the hypothesis. (lines 109-111)

“The study hypothesized that the two tested preventive programs reduced rotator cuff imbalances in competitive swimmers. Conversely, the absence of a preventive program does not affect rotator cuff strength and balance.”
